# HAODiff: Human-Aware One-Step Diffusion via Dual-Prompt Guidance

**Jue Gong**[1]*, **Tingyu Yang**[1]*, **Jingkai Wang**[1], **Zheng Chen**[1],
**Xing Liu**[2], **Hong Gu**[2], **Yulun Zhang**[1]†, **Xiaokang Yang**[1]
[1]Shanghai Jiao Tong University     [2]vivo Mobile Communication Co., Ltd

## Abstract

Human-centered images often suffer from severe generic degradation during transmission and are prone to human motion blur (HMB), making restoration challenging. Existing research lacks sufficient focus on these issues, as both problems often coexist in practice. To address this, we design a degradation pipeline that simulates the coexistence of HMB and generic noise, generating synthetic degraded data to train our proposed HAODiff, a human-aware one-step diffusion. Specifically, we propose a triple-branch dual-prompt guidance (DPG), which leverages high-quality images, residual noise (LQ minus HQ), and HMB segmentation masks as training targets. It produces a positive–negative prompt pair for classifier-free guidance (CFG) in a single diffusion step. The resulting adaptive dual prompts let HAODiff exploit CFG more effectively, boosting robustness against diverse degradations. For fair evaluation, we introduce MPII-Test, a benchmark rich in combined noise and HMB cases. Extensive experiments show that our HAODiff surpasses existing state-of-the-art (SOTA) methods in terms of both quantitative metrics and visual quality on synthetic and real-world datasets, including our introduced MPII-Test. Code is available at: https://github.com/gobunu/HAODiff.

## 1    Introduction

Human body restoration (HBR) focuses on recovering high-quality (HQ) images from low-quality (LQ) inputs that contain human images. When human subjects appear prominently in an image, it naturally attracts more viewer attention. However, real-world images frequently undergo degradation during capture or transmission, including human motion blur, noise, resolution loss, and JPEG compression artifacts. These distortions significantly hinder the recognition of human activities and limit the usefulness of such images in broader applications. As a result, many downstream tasks related to humans are negatively affected, such as 3D reconstruction [46, 64], human pose estimation [41, 71], and human-object interaction detection [53, 31].

To achieve better performance in practical scenarios, current blind image restoration (BIR) models typically rely on a large number of paired LQ and HQ images to accurately learn the complex mapping between LQ and HQ domains. Models in the HBR field follow the same principle. However, collecting a large number of real LQ-HQ image pairs is challenging. Real-world LQ images often undergo various unknown degradations, which may occur during transmission or even at the time of capture. Therefore, BIR models usually adopt a degradation pipeline to simulate real-world LQ conditions. This strategy allows models to fully leverage large-scale high-quality image restoration datasets, such as LSDIR [27], FFHQ [18], and PERSONA [11]. In the context of HBR, existing approaches [11, 70] generally use generic degradation types, including downsampling, compression, noise, and low-pass blur. A commonly adopted pipeline is from Real-ESRGAN [51], which is originally developed for blind super-resolution tasks. However, this pipeline may fail to cover the diverse degradation types that frequently occur in human-centric images.

---

*Equal contribution.

†Correspondence author: Yulun Zhang, yulun100@gmail.com.

39th Conference on Neural Information Processing Systems (NeurIPS 2025).

Among the various degradation types in human-centric images, human motion blur is particularly prevalent, yet it remains significantly underrepresented in existing degradation pipelines. Motion blur can be broadly categorized into two types. The first category is global motion blur, typically caused by camera movement during exposure. The second is local motion blur, caused by the rapid movement of objects in the scene. In human images, local motion blur mainly arises from human movement, known as human motion blur (HMB). Extensive research is conducted on the removal of global motion blur. For example, DeblurGAN [22] not only proposes an effective model for motion deblurring, but also provides a simulation pipeline that can generate realistic global motion blur. This enables training with large-scale synthetic data. In contrast, the restoration of local motion blur

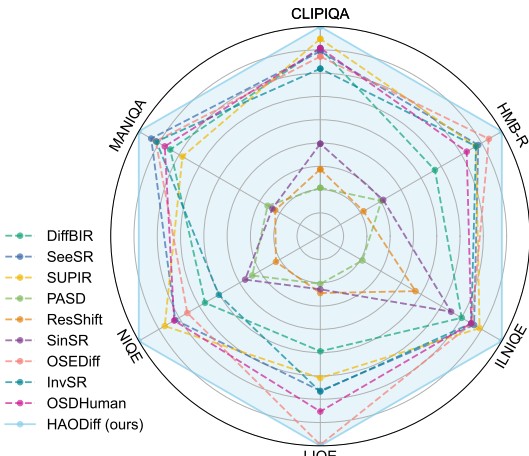

Figure 1: Performance comparison on introduced MPII-Test. HMB-R denotes the ratio of human motion blur detection instances before and after restoration. Lower-is-better metrics are inverted.

is typically based on real datasets, with ReLoBlur [24] being a representative example that covers diverse object motion. However, these datasets often focus on general objects and pay less attention to human-specific motion blur. Moreover, HMB in real-world human images often co-occurs with other degradation types. Existing deblurring models are usually designed to handle a single degradation, which limits their applicability in more complex, realistic scenarios.

Multiple factors, including training data and degradation pipelines, influence the quality of image restoration. Among them, the model architecture plays a critical role. In BIR tasks, models require strong generative capabilities to reconstruct damaged regions. Generative adversarial networks (GANs) [12] are considered one of the major starting points for modern image generation models. They have led to the development of many powerful restoration methods [65, 22, 51]. However, GANs often suffer from training instability and difficulties in controlling the generation process. Diffusion models and latent diffusion models (LDMs) [39] have further improved the framework for both generation and restoration. Some models [29, 59, 55, 62] use multi-step diffusion processes to restore high-quality images from heavily degraded inputs. Recently, one-step diffusion models [54, 49, 52, 61] also demonstrate strong BIR performance while significantly reducing resource consumption compared to multi-step diffusion. These models commonly extract object features from images and convert them into text or image embeddings. They are used as positive prompts to guide the model toward faithful restoration. To better leverage the guidance capabilities of text-to-image (T2I) foundation models, some methods [63, 60] introduce negative prompts using classifier-free guidance (CFG) [15]. These prompts help steer the model away from undesired content. However, these methods adopt fixed negative prompts, which limit their guidance effectiveness.

To address these limitations, we propose HAODiff, a novel one-step diffusion for human body restoration (HBR) that integrates CFG via dual-prompt guidance (DPG). **Firstly**, to compensate for the lack of human motion blur (HMB) in existing pipelines, we propose a new one. In the preprocessing stage, we apply a human body-part segmentation model to the HQ training images to obtain part-specific masks. These masks are then combined with a motion blur simulation module to synthesize HMB. Integrated into a generic two-stage pipeline, our approach allows the inclusion of HMB. **Secondly**, we design a triple-branch dual-prompt guidance, named DPG, which is based on the Swin Transformer [33]. During training with the degradation pipeline, one branch of DPG predicts the HQ image to produce the positive prompt. The other two are used to predict the residual noise (LQ minus HQ) and the human motion blur segmentation mask, both serving as sources for the negative prompt. **Thirdly**, we propose a human-aware one-step diffusion via dual-prompt guidance, named HAODiff. The model accepts both positive and negative prompts generated by DPG. With the CFG strategy, the negative prompt replaces the unconditional input and guides the model to learn how to approach HQ features while avoiding LQ characteristics. **In addition**, to further evaluate performance under both generic and HMB degradations, we construct a new benchmark, MPII-Test. It is curated from the MPII Human Pose dataset [1] and contains 5,427 real-world degraded human images, many of which include rich motion blur patterns. As shown in Fig. 1, the results demonstrate our model's strong ability to restore human images with both HMB and other degradations.

In summary, we make the following four key contributions:

- We propose a new degradation pipeline for human body restoration. It explicitly incorporates human motion blur into the degradation process, enabling the model to learn from more realistic degradation scenarios and improving its generalization in real-world applications.

- We propose HAODiff, a one-step diffusion model that integrates the classifier-free guidance strategy. By replacing fixed input with adaptive negative prompts, the model is guided toward HQ features and away from LQ signals, enhancing robustness to noise.

- We design a dual-prompt guidance (DPG), which efficiently generates distinct positive and negative prompts for each LQ image. This addresses the challenge of constructing targeted negative prompts and provides spatial guidance on the location of human motion blur.

- Our proposed method, HAODiff, achieves significant state-of-the-art performance on both existing test datasets and our newly introduced benchmark. It delivers strong visual quality and competitive quantitative results while maintaining high computational efficiency.

## 2 Related Work

### 2.1 Human Body Restoration

Human body restoration (HBR) constitutes a specialized subfield that applies blind image restoration (BIR) techniques specifically to the restoration of LQ human images. Contemporary mainstream BIR methods [28, 51, 65, 29] show remarkable efficacy across a wide range of natural scenarios. However, their direct application to human body images frequently causes joint misalignment and limb distortion. Previous models [32, 50, 70, 11] have addressed this task with varying methodologies. Among them, DiffBody [70] pioneers the application of diffusion model in body-region image enhancement under the guidance of the body attention module. A notable contribution to the field is OSDHuman [11], which proposes a one-step diffusion model for human body restoration using a high-fidelity image embedder while introducing the PERSONA benchmark dataset.

Human motion blur represents one of the most challenging degradation modalities in HBR contexts. Current deep-learning-based deblurring research primarily focuses on objects and scenes [36, 6, 45, 22, 3]. DeblurGAN [22] represents the inaugural application of a conditional adversarial network to blind motion deblurring and introduced a random-trajectory simulation pipeline for synthesizing motion-blurred data. More recently, OSDD [30], a one-step diffusion model for motion deblurring that substantially enhances the computational efficiency of diffusion-based restoration. Moreover, ReLoBlur [24] provides the first real-world locally blurred dataset captured with synchronized light-field cameras, while developing a Blur-Aware Gating network to restore these regions.

### 2.2 Diffusion Models

Text-to-image (T2I) diffusion models are repurposed for image restoration tasks due to their powerful prior knowledge in image generation [48, 29, 55, 60, 59]. For instance, Stable Diffusion (SD) [43], with scalable networks and controllable generation, demonstrates the capability to inject vivid details into LQ images. Building on this foundation, StableSR [48] enhances image restoration via a fine-tuned time-aware encoder and progressive sampling strategies. By implementing a degradation-aware prompt extractor, SeeSR [55] guides diffusion models to generate semantically accurate HQ images with precise prompt control. DiffBIR [56] first employs an initial restoration module before incorporating SD for detail refinement. Furthermore, SUPIR [60] leverages SDXL [38] as its prior, achieving impressive results through high-quality datasets and innovative positive-negative sample strategies. However, their adherence to conventional T2I diffusion paradigms necessitates numerous sampling steps, resulting in computational inefficiency and excessive parametric complexity.

Contemporary research increasingly focuses on addressing the inefficiencies of multi-step diffusion processes in BIR tasks by leveraging more efficient one-step diffusion [52, 54, 63, 61]. SinSR [52] advances this direction by distilling a deterministic mapping from a teacher diffusion model. OSEDiff [54] injects LQ images into the latent space as the diffusion starting point and employs variational score distillation to align with the image prior of SD. More recently, based on diffusion inversion, InvSR [61] uses noise prediction to create optimal intermediate sampling states. Nevertheless, they insufficiently exploit the semantic information inherent within the LQ images themselves. The global semantic content of LQ images can be effectively processed and utilized as prompt embeddings for diffusion models, substantially enhancing one-step diffusion restoration capabilities.

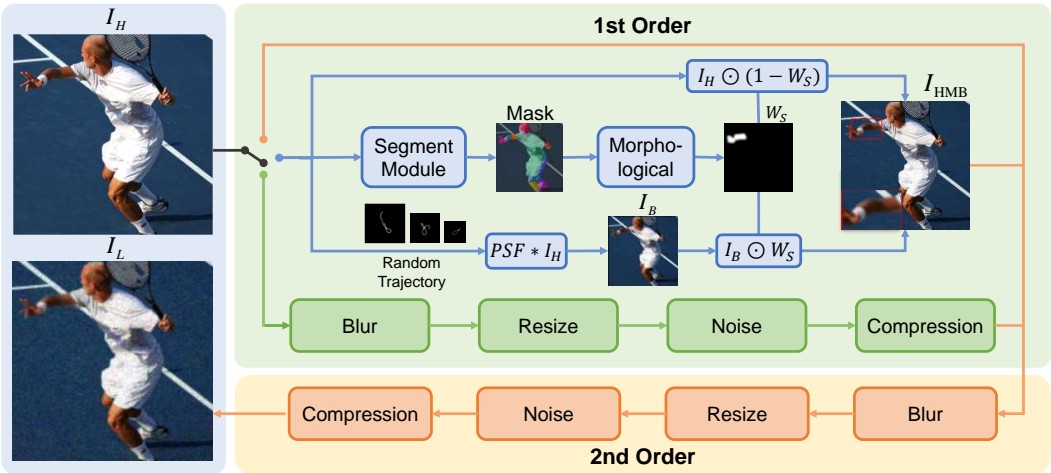

Figure 2: Degradation pipeline overview. The first order contains three possible cases: (i) no degradation, (ii) human motion blur (HMB), and (iii) generic degradation. The HMB branch conducts body-part segmentation to obtain masks, then morphs them to yield the spatial weight map $W_s$. This map is applied to the motion blur image $I_B$, which is generated by convolving the clean image $I_H$ with a point spread function (PSF) derived from a random trajectory. The result is combined with $I_H$ to create the synthetic HMB image $I_{HMB}$. The second applies conventional generic degradation.

## 3 Method

### 3.1 Degradation Pipeline with Human Motion Blur (HMB)

Obtaining paired training data where degradation incorporates HMB remains a significant challenge. Prior works [36, 37, 24] construct global or local motion blur by capturing high-quality continuous frames from high frame-rate videos and averaging them to synthesize motion blur. However, this strategy struggles to cover the diversity of real-world human activities, limiting the generalizability of the dataset. Another method involves convolving natural images with blur kernels generated from complex motion trajectories [4, 44, 57, 2], as exemplified by DeblurGAN [22]. Building upon these methods, we design a degradation pipeline that simulates human motion blur and incorporates generic degradation processes based on Real-ESRGAN [51] to train for human body restoration.

The pipeline is illustrated in Fig. 2. In the first order of degradation, we use Sapiens [19] for body-part segmentation on HQ images $I_H$, yielding part masks. These masks are grouped into six categories (head, left/right upper limbs, left/right lower limbs, and the whole body), from which one category is randomly selected for motion blur simulation in subsequent steps. Since segmented regions do not inherently correspond to realistic motion patterns, we apply morphological operations, including erosion, dilation, and Gaussian blurring, to the selected masks. These processed masks are then normalized to generate a spatial weight map $W_s$, which can be formalized as:

$$W_s = (Norm \circ Morph \circ Seg)(I_H) \quad \text{with} \quad Morph = G \circ D \circ E, \tag{1}$$

where $Seg(\cdot)$ refers to the segmentation operation, and $Morph(\cdot)$ denotes morphological operations: Gaussian blur ($G$), dilation ($D$) and erosion ($E$). $Norm(\cdot)$ denotes scaling values to $[0, 1]$.

In parallel, we generate a globally blurred image using a strategy similar to DeblurGAN [22]: a continuous random trajectory is simulated via a Markov process and converted to a discrete point spread function (PSF) through bilinear interpolation. The PSF is then convolved with the HQ image via FFT (fast Fourier transform, denoted as $\mathcal{F}$) convolution to produce global motion blur. Finally, we blend the original and blurred images using the spatial weight map, yielding the HMB image $I_{HMB}$:

$$I_{HMB} = W_s \odot I_H + (1 - W_s) \odot I_B \quad \text{with} \quad I_B = \mathcal{F}^{-1}\left(\mathcal{F}(PSF) \odot \mathcal{F}(I_H)\right). \tag{2}$$

Notably, HMB is typically caused by the subject's movement during capture and should logically precede all other degradations in the simulation process. Therefore, to maintain logical consistency, we place HMB in the first-order stage. Since motion blur and severe degradations do not always occur, we define the first-order degradation as one of three possible conditions: no degradation, HMB, or generic degradation. After this stage, the image undergoes second-order degradation, which includes common types such as blur, resizing, noise, and compression. The generic degradations in both the first and second orders follow the strategy of Real-ESRGAN [51]. Upon traversing this comprehensive degradation pipeline, we obtain synthesized LQ images $I_L$ for subsequent processing.

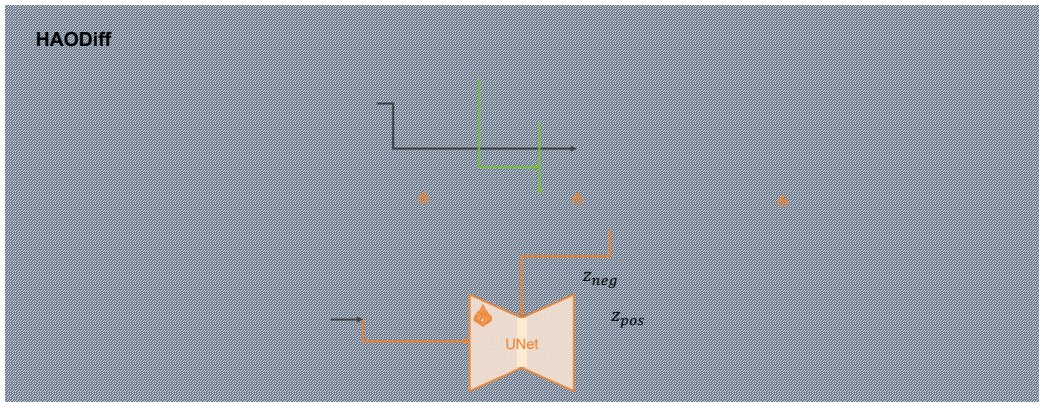

Figure 3: Model structure of our HAODiff. **Stage 1**: We train a triple-branch dual-prompt guidance (DPG). The core structure consists of downsampler and upsampler ($H_D$, $H_{Ui}$), as well as feature extraction and reconstruction modules ($H_E$ and $H_{Ri}$). Both $H_E$ and $H_{Ri}$ are composed of two residual Swin Transformer blocks (RSTB). The three branches are individually trained with the human motion blur segmentation masks ($M_{HMB}$), residual noise ($I_L - I_H$), and high-quality images ($I_H$). **Stage 2**: We leverage DPG combined with prompt embedder to provide positive and negative prompt pairs to the one-step diffusion (OSD) model. The UNet generates $z_{pos}$ and $z_{neg}$, used to obtain the predicted latent vector $\hat{z}_H$ through classifier-free guidance (CFG) and denoising operations.

## 3.2 Stage 1. Dual-Prompt Guidance (DPG)

A key challenge in prompt extraction is effectively predicting high-quality (HQ) features from low-quality (LQ) images. OSDFace [49] tackles this via HQ–LQ embedding alignment. However, alignment alone is insufficient because predicting HQ features from LQ involves restoration, which is challenging for prompt extractors. Directly training the prompt extractor with LQ inputs and HQ targets embeds HQ features within intermediate model layers. Additionally, for restoration models, defining the negative prompt is essential: the restored images should not completely diverge from the LQ input but rather specifically from the residual noise (LQ minus HQ), which contains degradation without structural information. Otherwise, the restored images may lack fidelity. Furthermore, local noise (*e.g.*, human motion blur) hidden in global noise is hard to locate and remove. Existing methods [24, 25] use segmentation supervision to locate motion blur regions. A dedicated module that supplies explicit positional information can further enhance noise localization and removal.

**DPG Structure.** To provide the model with image embeddings as prompts, the Vision Transformer [10, 33] framework is suitable. It can convert RGB images into embeddings for feature extraction. Furthermore, SwinIR [28] in image restoration, using composed residual Swin Transformer blocks (RSTB) to recover HQ from LQ, achieves great success. Inspired by SwinIR, our prompt extractor structure is shown in Fig. 3. First, we use a convolutional structure ($H_D$). It downsamples the LQ image ($I_L$) size 512×512 by 4. And it increases channels to match the sequence length by the Swin Transformer, which we set to 150. Next, the backbone network (denoted $H_E$) extracts features through two RSTBs, with six Swin Transformer layers (STL). Afterward, the model splits into three reconstruction branches (denoted as $H_{Ri}, i \in [1, 3]$). Each branch also contains two RSTBs but only three STLs. Each branch then passes through a convolutional upsampler (denoted as $H_{Ui}, i \in [1, 3]$). The three branches are designed to predict the HQ image ($\hat{I}_H^P$), the residual noise between LQ and HQ ($\hat{I}_R$), and the HMB segmentation mask ($\hat{M}_{HMB}$). This process can be formulated as follows:

$$(\hat{I}_H^P, \hat{I}_R, \hat{M}_{HMB}) = ((H_{Ui} \circ H_{Ri})(F))_{i=1}^3 \quad \text{with} \quad F = (H_E \circ H_D)(I_L). \quad (3)$$

**Training Objective of DPG.** The predicted HQ image ($\hat{I}_H^P$) and the residual noise between LQ and HQ ($\hat{I}_R$) are optimized jointly using the pixel-wise L1 loss $\mathcal{L}_1$. The HMB-aware branch, responsible for predicting the human motion blur (HMB) segmentation mask ($\hat{M}_{HMB}$), uses the Dice [8] loss $\mathcal{L}_{Dice}$. The overall training objective can be formulated as follows:

$$\mathcal{L} = \mathcal{L}_1(\hat{I}_H^P, I_H) + \mathcal{L}_1\left(\hat{I}_R, (I_L - I_H)\right) + \alpha \cdot \mathcal{L}_{Dice}(\hat{M}_{HMB}, M_{HMB}), \quad (4)$$

where $M_{HMB}$ is obtained by binarizing $W_s$ from Sec. 3.1. The Dice loss is defined as:

$$\mathcal{L}_{Dice}(\hat{M}_{HMB}, M_{HMB}) = 1 - \frac{2 \cdot |\hat{M}_{HMB} \cap M_{HMB}|}{|\hat{M}_{HMB}| + |M_{HMB}|} = 1 - \frac{2 \cdot \text{sum}(\hat{M}_{HMB} \odot M_{HMB})}{\text{sum}(\hat{M}_{HMB}) + \text{sum}(M_{HMB})}. \quad (5)$$

### 3.3 Stage 2. One-Step Diffusion (OSD) Model

**Model Structure.** The latent diffusion model (LDM) [39] adds noise in latent space at timestep $t$ as $z_t = \sqrt{\bar{\alpha}_t}\, z + \sqrt{1 - \bar{\alpha}_t}\, \varepsilon$, where $\varepsilon \sim \mathcal{N}(0, I)$ and $\bar{\alpha}_t$ denotes the cumulative product of $\alpha_s$ up to timestep $t$: $\bar{\alpha}_t = \prod_{s=1}^{t}(1 - \beta_s)$. The reverse process predicts noise with parameter $\theta$. During inference, the LDM employs the DDIM [42] to accelerate the reverse process, simplified as follows:

$$z_{t-1} = \sqrt{\bar{\alpha}_{t-1}} \left( \frac{z_t - \sqrt{1 - \bar{\alpha}_t}\, \varepsilon_\theta(z_t; p, t)}{\sqrt{\bar{\alpha}_t}} \right) + \sqrt{1 - \bar{\alpha}_{t-1} - \sigma_t^2}\, \varepsilon_\theta(z_t; p, t) + \sigma_t \varepsilon, \quad (6)$$

where $t - 1$ represents the next step in the DDIM sampling sequence. The $\sigma_t^2$ is defined as follows:

$$\sigma_t^2 = \eta^2 \frac{1 - \bar{\alpha}_{t-1}}{1 - \bar{\alpha}_t} \left( 1 - \frac{\bar{\alpha}_t}{\bar{\alpha}_{t-1}} \right), \; \eta \in [0, 1]. \quad (7)$$

In restoration tasks, for stability, we set $\eta=0$, so $\sigma_t=0$. With a specific timestep $\tau$, the latent vector $z_\tau$ should correspond to the noisy representation $z_L$ encoded from the LQ image $I_L$ by the variational autoencoder (VAE) [21] encoder $E_\theta$. $t - 1$ should correspond to step 0 in OSD. Thus $\bar{\alpha}_{t-1} = 1$, since the output is the noise-free latent vector $z_0$, also denoted as $\hat{z}_H$: $\hat{z}_H = (z_t - \sqrt{1 - \bar{\alpha}_t}\, z_\varepsilon)/\sqrt{\bar{\alpha}_t}$. When the classifier-free guidance (CFG) [15] is employed, the predicted noise $z_\varepsilon$ is defined as:

$$z_\varepsilon = z_{\text{neg}} + \lambda_{\text{cfg}} \cdot (z_{\text{pos}} - z_{\text{neg}}) \quad \text{with} \quad z_{\text{pos}} = \varepsilon_\theta(z_L; p_{\text{pos}}, \tau),\; z_{\text{neg}} = \varepsilon_\theta(z_L; p_{\text{neg}}, \tau) \quad (8)$$

where $p_{\text{pos}}$ and $p_{\text{neg}}$ denote the positive and negative prompts provided to the model. The parameter $\lambda_{\text{cfg}}$ controls the intensity of CFG, balancing the influence between dual predictions. The resulting latent vector $\hat{z}_H$ is then passed through the VAE decoder $D_\theta$ to obtain the restored image $\hat{I}_H^D$.

As for the prompt extractor part, after training the dual-prompt guidance (DPG) in Sec. 3.2, we further adapt it to be compatible with providing prompt embeddings. We extract the output feature from the last layer of the $H_{Ri}$ to obtain embedding features with the most informative representation. However, the size of this output significantly differs from the embedding size required by SD, and its feature space distribution also deviates from standard text embeddings. Therefore, before integrating into Stable Diffusion (SD), we apply a nonlinear mapping and feature compression using a linear complexity Performer [7] Encoder and an attention pooling [23] module, denoted prompt embedder, as shown in Fig. 4. The features of the two negative branches are concatenated and, with the positive branch, fed into different prompt embedders. This process yields dual-prompt embeddings $p_{\text{pos}}$ and $p_{\text{neg}}$. Finally, by concatenating these embeddings along the batch size dimension, the receiving UNet [40] can efficiently obtain both $z_{\text{pos}}$ and $z_{\text{neg}}$ in parallel.

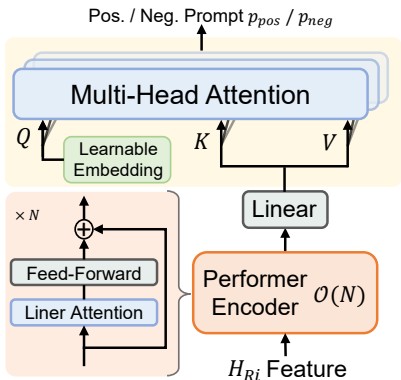

Figure 4: Structure of the prompt embedder. The Attention Pooling uses a learnable embedding as $Q$, while $K$ and $V$ from the output of Performer Encoder, whose depth $N$ is set to 6.

**Training Objective.** The human body restoration model aims to recover high-quality human images with rich details from degraded inputs. During training, we use pixel-level mean squared error loss $\mathcal{L}_{\text{MSE}}$ to minimize reconstruction errors. To enhance edge responses, we also employ an edge-aware DISTS perceptual loss, denoted as $\mathcal{L}_{\text{EA}}$. It is calculated by feeding both the original image and its edge-enhanced version, obtained using the Sobel operator $\mathcal{S}(\cdot)$, into the DISTS [9] function:

$$\mathcal{L}_{\text{EA}} = \mathcal{L}_{\text{dists}}(\hat{I}_H, I_H) + \mathcal{L}_{\text{dists}}(\mathcal{S}(\hat{I}_H), \mathcal{S}(I_H)). \quad (9)$$

Previous study [63] shows that even when the restored image is similar to the original, distortions may still occur in the latent vector distribution. To address this, we utilize a pre-trained SD UNet downsampling module $\mathcal{D}_\psi$ as a discriminator and calculate the generator loss $\mathcal{L}_{\mathcal{G}}$. It helps the model learn a more realistic data distribution. The total loss function $\mathcal{L}_{\text{total}}$ is defined as follows:

$$\mathcal{L}_{\text{total}} = \mathcal{L}_{\text{MSE}}(\hat{I}_H, I_H) + \mathcal{L}_{\text{EA}}(\hat{I}_H, I_H) + \beta \cdot \mathcal{L}_{\mathcal{G}}(\hat{z}_H). \quad (10)$$

The generative adversarial network (GAN) [12] loss consists of the generator loss $\mathcal{L}_{\mathcal{G}}$ and the discriminator loss $\mathcal{L}_{\mathcal{D}}$. Following previous works [63, 49], we define them as follows:

$$\begin{aligned} \mathcal{L}_{\mathcal{G}}(\hat{z}_H) &= -\mathbb{E}_t \left[ \log \mathcal{D}_\psi \left( F(\hat{z}_H, t) \right) \right], \\ \mathcal{L}_{\mathcal{D}}(\hat{z}_H, z_H) &= -\mathbb{E}_t \left[ \log \left( 1 - \mathcal{D}_\psi \left( F(\hat{z}_H, t) \right) \right) \right] - \mathbb{E}_t \left[ \log \mathcal{D}_\psi \left( F(z_H, t) \right) \right], \end{aligned} \quad (11)$$

here, $z_H$ represents the latent vector of a high-quality human image $I_H$, and $F(\cdot)$ denotes the diffusion noise addition process, which is related to a randomly chosen timestep $t \in [0, T]$.

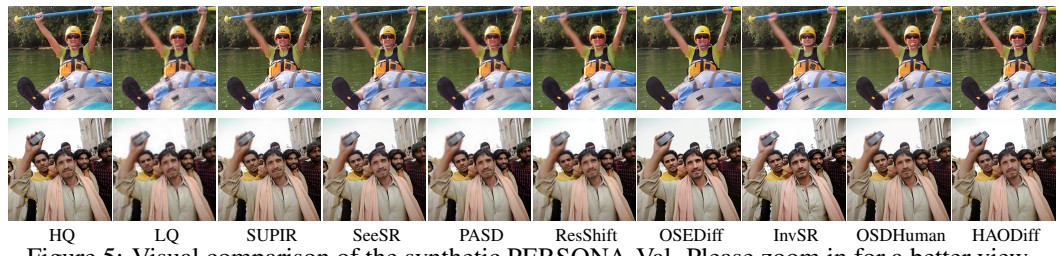

| | HQ | LQ | SUPIR | SeeSR | PASD | ResShift | OSEDiff | InvSR | OSDHuman | HAODiff |

Figure 5: Visual comparison of the synthetic PERSONA-Val. Please zoom in for a better view.

| Methods | Step | Time (s) | DISTS↓ | LPIPS↓ | TOPIQ↑ | FID↓ | C-IQA↑ | M-IQA↑ | NIQE↓ | LIQE↑ | IL-NIQE↓ |
|---|---|---|---|---|---|---|---|---|---|---|---|
| SUPIR [60] | 50 | 26.67 | 0.1415 | 0.2929 | 0.4080 | 13.8357 | 0.7908 | 0.6811 | 3.7769 | 4.2303 | 21.3221 |
| DiffBIR [29] | 50 | 9.03 | 0.1402 | 0.2797 | 0.4319 | 12.9306 | 0.7792 | 0.6965 | 3.9404 | 4.4211 | 21.4859 |
| SeeSR [55] | 50 | 5.05 | 0.1295 | 0.2555 | 0.4476 | 12.8225 | 0.7620 | 0.6979 | 3.5744 | 4.5722 | 21.1563 |
| PASD [59] | 20 | 3.15 | 0.1469 | 0.2910 | 0.4383 | 15.6758 | 0.6121 | 0.6477 | 3.8042 | 4.0878 | 25.4919 |
| ResShift [62] | 15 | 2.88 | 0.1638 | 0.2848 | 0.4237 | 15.6502 | 0.5365 | 0.5650 | 5.2973 | 3.3929 | 25.3706 |
| SinSR [52] | 1 | 0.19 | 0.1579 | 0.2840 | 0.4037 | 16.9203 | 0.6037 | 0.5802 | 4.4221 | 3.6340 | 21.1563 |
| OSEDiff [54] | 1 | 0.13 | 0.1340 | 0.2507 | 0.4539 | 14.7489 | 0.6961 | 0.6769 | 3.4689 | 4.7567 | 21.9784 |
| InvSR [61] | 1 | 0.17 | 0.1424 | 0.2709 | 0.4297 | 13.2950 | 0.6886 | 0.6845 | 3.7446 | 4.3290 | 21.6265 |
| OSDHuman [11] | 1 | 0.11 | 0.1356 | 0.2384 | 0.4680 | 14.4121 | 0.7312 | 0.6908 | 3.8278 | 4.7512 | 22.3524 |
| HAODiff (ours) | 1 | 0.20 | 0.1023 | 0.2046 | 0.5161 | 8.3623 | 0.7737 | 0.7097 | 2.8298 | 4.8485 | 18.5986 |

Table 1: Quantitative results on PERSONA-Val and inference time comparison. The top two scores are highlighted in red and cyan for all methods. C-IQA stands for CLIPIQA, and M-IQA stands for MANIQA. Inference is performed on images of 512×512 resolution on NVIDIA RTX A6000.

## 4 Experiments

### 4.1 Experimental Settings

**Training and Testing Datasets.** Our model is trained on the PERSONA [11], with 20k images sampled from both LSDIR [27] and FFHQ [18]. We randomly crop LSDIR to 512×512, and pre-downsample FFHQ to the same size. We generate synthetic HQ-LQ image pairs using our degradation pipeline. For testing, we use PERSONA-Val and PERSONA-Test from OSDHuman [11]. Additionally, we select images from the MPII Human Pose dataset [1] using the data selection pipeline from OSDHuman, excluding the quality filtering stage. To accommodate the bounding boxes used in this process, we extend the outermost annotated key points outward by a certain margin to form enclosing rectangles, which are used as bounding boxes. This process yields MPII-Test, which consists of 5,427 real-world images with diverse human motion blur (HMB). Using our degradation pipeline, we fine-tune YOLO11 [17] to detect HMB, finding 1,765 instances in 1,326 images.

**Evaluation Metrics.** For PERSONA-Val, we use both full-reference and no-reference metrics. For full-reference perceptual quality assessment, we use DISTS [9], LPIPS [68], and TOPIQ [5]. Additionally, we calculate FID [14] to measure the distribution distance between the restored images and ground truth. For no-reference metrics, we use CLIPIQA [47], NIQE [66], MANIQA-pipal [58], LIQE [69], and IL-NIQE [67]. These no-reference metrics are also applied to PERSONA-Test and MPII-Test. Additionally, for MPII-Test, we use the fine-tuned YOLO model to detect HMB instances and calculate the ratio of detected instances in restored images to those in original images, denoted as HMB-R. The fine-tuned YOLO detector is evaluated on a dedicated hold-out set containing 4,216 images, achieving an mAP@0.5 of 0.6223 and a precision of 0.6738.

**Implementation Details.** In stage 1, the DPG training process balances the magnitude of the L1 loss and Dice loss by setting the $\alpha$ in Eq. (4) to $2\times10^{-2}$. We use the Adam optimizer [20] with learning rate $2\times10^{-3}$ and batch size 16. The STLs in $H_E$ have 6 heads, while those in $H_{Ri}$ use 3 heads. For HBR segmentation, the third branch outputs a single channel and utilizes the sigmoid activation function, while the other two output three channels without the activation function. The training is conducted for 20k iterations on 4 NVIDIA RTX A6000 GPUs. In stage 2, we set $\beta$ in Eq. (10) to $1\times10^{-2}$ and use the pretrained SDXL [38] UNet as the discriminator following D$^3$SR [26]. The $\lambda_{\text{cfg}}$ in Eq. (8) is set to 3.5. The AdamW optimizer [34] used in stage 2 has learning rate $1\times10^{-5}$ and batch size 2. The base model is SD2.1-base [43] and LoRA [16] is used to train the UNet with a LoRA rank 16. The training is conducted for 120k iterations on 2 NVIDIA RTX A6000 GPUs.

**Compared State-of-the-Art (SOTA) Methods.** We compare HAODiff with multi-step diffusion models, including DiffBIR [29], SeeSR [55], SUPIR [60], PASD [59] and ResShift [62]; and one-step diffusion models, including SinSR [52], OSEDiff [54], InvSR [61], and OSDHuman [11].

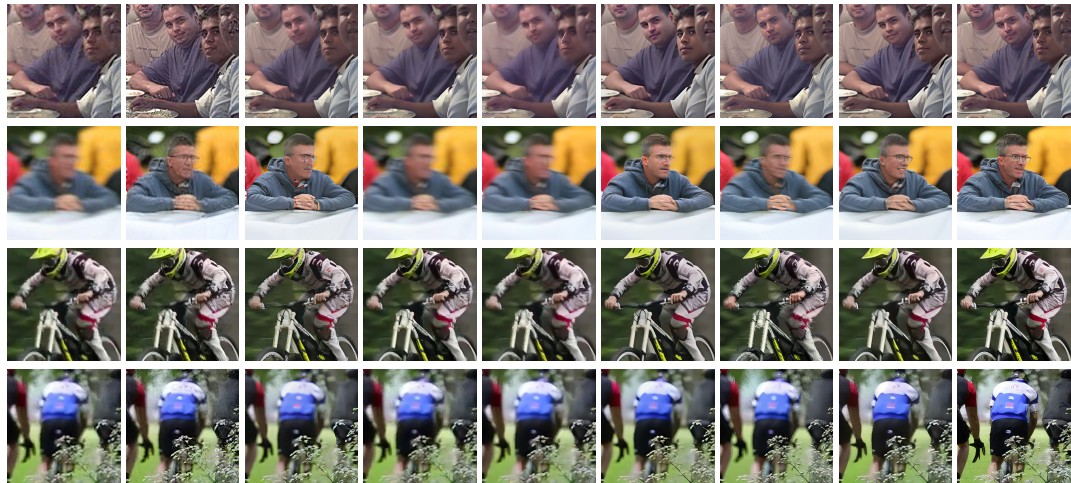

| | LQ | SUPIR | SeeSR | PASD | ResShift | OSDiff | InvSR | OSDHuman | HAODiff |

Figure 6: Visual comparison of the real-world PERSONA-Test and MPII-Test datasets in challenging and representative cases. Please zoom in for a better view.

| Methods | PERSONA-Test | | | | | MPII-Test | | | | | |
|---|---|---|---|---|---|---|---|---|---|---|---|
| | C-IQA↑ | M-IQA↑ | NIQE↓ | LIQE↑ | IL-NIQE↓ | C-IQA↑ | M-IQA↑ | NIQE↓ | LIQE↑ | IL-NIQE↓ | HMB-R↓ |
| SUPIR [60] | 0.7106 | 0.6798 | 4.0734 | 4.0501 | 22.6182 | 0.6702 | 0.6256 | 4.4231 | 3.4295 | 26.0135 | 0.2776 |
| DiffBIR [29] | 0.7287 | 0.6812 | 4.9820 | 3.8870 | 25.4183 | 0.6531 | 0.6405 | 5.1638 | 3.1343 | 27.8964 | 0.5705 |
| SeeSR [55] | 0.6968 | 0.6759 | 4.1126 | 4.0800 | 23.4376 | 0.6478 | 0.6636 | 4.6152 | 3.5709 | 26.5874 | 0.2612 |
| PASD [59] | 0.5765 | 0.6703 | 3.8728 | 3.8901 | 24.9226 | 0.4023 | 0.5220 | 6.0325 | 2.3903 | 38.3461 | 0.9518 |
| ResShift [62] | 0.5544 | 0.6101 | 4.8438 | 3.4981 | 25.0764 | 0.4356 | 0.5128 | 6.4687 | 2.4915 | 32.7579 | 1.0799 |
| SinSR [52] | 0.5882 | 0.6010 | 4.7510 | 3.5339 | 23.3862 | 0.4816 | 0.5162 | 5.9015 | 2.4504 | 29.0120 | 0.9394 |
| OSEDiff [54] | 0.6734 | 0.6919 | 4.4600 | 4.4296 | 24.4183 | 0.6385 | 0.6580 | 4.8383 | 4.1664 | 26.8298 | 0.1853 |
| InvSR [61] | 0.6837 | 0.7122 | 4.1694 | 4.2582 | 22.8932 | 0.6166 | 0.6573 | 5.4187 | 3.5793 | 26.8856 | 0.2771 |
| OSDHuman [11] | 0.7155 | 0.6977 | 4.1287 | 4.3202 | 24.8712 | 0.6537 | 0.6471 | 4.5960 | 3.7991 | 26.8940 | 0.3433 |
| HAODiff (ours) | 0.7210 | 0.7057 | 3.8269 | 4.2375 | 21.9784 | 0.6923 | 0.6787 | 3.9450 | 4.1777 | 23.6714 | 0.0929 |

Table 2: Quantitative comparison on real-world benchmarks, with top two results respectively highlighted in red and cyan. HMB-R indicates the ratio of human motion blur instances compared with restored and original images. C-IQA stands for CLIPIQA, and M-IQA stands for MANIQA.

## 4.2 Main Results

**Quantitative Comparisons.** Table 1 presents quantitative results on the synthesized PERSONA-Val. Despite HAODiff's overwhelming inference speed advantage over multi-step diffusion models and comparable performance to some one-step diffusion (OSD) models, it achieves top scores across all full-reference metrics. In no-reference evaluation, it ranks first in NIQE and IL-NIQE, indicating natural image synthesis, and leads in MANIQA and LIQE, reflecting alignment with human aesthetics. Among OSD models, HAODiff exhibits the best performance in CLIPIQA, showcasing high-quality restoration. Table 2 compares HAODiff on PERSONA-Test and MPII-Test. On PERSONA-Test, HAODiff maintains superior performance, leading on most metrics. For the human motion blur (HMB)-rich MPII-Test, HAODiff tops every metric. Meanwhile, our model achieved the lowest HMB-R, demonstrating strong HBR performance and effective restoration of HMB content.

**Qualitative Comparisons.** Visual comparisons with SOTA methods are shown in Figs. 5 and 6. HAODiff effectively restores generic degradations in low-quality (LQ) images, producing visually clear results. On the LQ human body dataset PERSONA-Test, our model achieves notably better reconstruction of both body and facial regions. Specifically, it is capable of restoring the applied HMB. On PERSONA-Val and MPII-Test, HAODiff shows clear advantages in handling human images with prominent HMB artifacts, delivering effective restoration. In regions affected by severe motion blur, the model produces results with sharp contours and realistic visual quality. Compared to both one-step and multi-step diffusion models, HAODiff consistently generates more detailed and natural results. It avoids the excessive smoothness observed in PASD [59], ResShift [62], and OSEDiff [54], as well as the over-sharpened and distorted textures found in SUPIR [60] and InvSR [61]. In addition, HAODiff produces more realistic clothing details, accurately reflecting fabric textures and folds under natural lighting conditions. Moreover, HAODiff specifically restores blurred human regions without altering the depth of field, conforming to natural optical factors, thereby preventing unrealistic texture synthesis in the background. Further visual comparisons are provided in the supplementary material.

## 4.3 Ablation Study

**HMB-aware Branch of DPG.** To assess the impact of the human motion blur (HMB)-aware branch (the third branch) in addressing HMB, we conducted a targeted ablation study. Specifically, we train HAODiff using a version of DPG that excludes the HMB-aware branch while retaining the other two branches. The training is performed for the same number of epochs, with identical settings and CFG strategy, and the model is further evaluated on the HMB-rich MPII-Test. The results are shown in Fig. 7. Without the guidance of the HMB-aware branch, the model shows a clear decline in its ability to recover the HMB regions. Specifically, the arms of the subject in the LQ image suffer from severe motion blur; without the branch, the model removes only non-motion-blur noise, whereas with the branch the arms are sharply restored. Quantitative comparisons between the two configurations are also provided, with LIQE and HMB-R scores annotated below each strategy. These results indicate that incorporating the HMB-aware branch not only preserves the overall restoration quality but also significantly improves the model's ability to handle HMB-specific degradation.

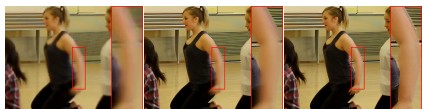

| LQ | w/o 3rd branch | w/ 3rd branch |
|---|---|---|
| LIQE↑ / HMB-R↓ | 4.1015 / 0.2816 | **4.1777 / 0.0929** |

Figure 7: Comparison of whether to leverage the third (HMB-aware) branch.

**The Effectiveness of DPG.** In order to validate the effectiveness of DPG, we compare it against several prompt-based guidance strategies, including: the fixed positive–negative prompt pair with CFG following S3Diff [63], the degradation aware prompt extractor (DAPE) from OSEDiff [54], and the high-fidelity image embedder (HFIE) from OSDHuman [11].

| Prompt Methods | DISTS↓ | MS-SWD↓ | FID↓ | CLIPIQA↑ | NIQE↓ |
|---|---|---|---|---|---|
| Text Pair [63] | 0.1056 | 0.3815 | 8.5215 | 0.7648 | 2.9692 |
| DAPE [54] | 0.1047 | 0.4296 | 8.6143 | 0.7693 | 3.0779 |
| HFIE [11] | 0.1060 | 0.4066 | 9.2570 | 0.7681 | 3.2063 |
| **DPG (ours)** | **0.1023** | **0.2829** | **8.3623** | **0.7737** | **2.8298** |

Table 3: Comparison of different prompt methods. Text Pair represents using a fixed pos-neg prompt pair with CFG.

Table 3 reports quantitative results on PERSONA-Val, where DPG consistently delivers superior restoration quality across multiple metrics. Moreover, we observe that other methods tend to produce more pronounced color shifts compared to DPG-guided outputs. These color shifts significantly affect human perception of image fidelity. And once they occur, it is challenging to eliminate them through post-processing methods (*e.g.*, wavelet-based correction [35]). To quantify this effect and assess the impact on model stability, we employ MS-SWD [13], a metric for measuring color distribution differences between the processed results and reference images. With HQ images as reference, the findings clearly show that DPG yields the smallest color discrepancies. DPG's superior performance in this respect stems from its dual-prompt design: the negative path captures residual noise, implicitly including color-shift degradation. During training, DPG more accurately anchors the original color distribution and detects color drift in the noise. This capability is then explicitly reinforced through classifier-free guidance (CFG), thereby resulting in higher fidelity and robustness.

## 5 Limitation and Conclusion

**Limitation.** Images with heavy global degradation and strong local motion blur, particularly in regions with fast-moving limbs, remain challenging for all methods, including ours. A few samples of this type appear in the MPII-Test. Although HAODiff outperforms all baselines (many of them fail completely, as shown in Fig. 8 and supplementary material), our results remain imper-

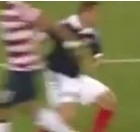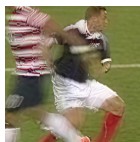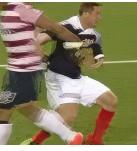

| LQ | SUPIR [60] | HAODiff (ours) |
|---|---|---|

Figure 8: Challenge task from MPII-Test.

fect. Limb poses may appear unnatural, and overlapping objects may be misrestored. These limitations prompt us to reconsider the boundary between pure restoration and content generation. **Moreover**, similar to common CFG strategies in image restoration, our method employs UNet to compute the latent vector twice. These two passes thus incur a modest efficiency penalty compared to OSEDiff [54] and OSDHuman [11] (see Tab. 1). In future work, we will develop more efficient CFG strategies and explore approaches to handle such extreme real-world cases, recover finer details.

**Conclusion.** We propose HAODiff, a human-aware one-step diffusion model. It perceives and addresses degradations around humans (*e.g.*, human motion blur). The restored images are perceptually aligned with human vision, minimizing color shifts while maintaining high quality. Our model achieves these results by leveraging a degradation pipeline with human motion blur and a triple-branch dual-prompt guidance (DPG). The pipeline simulates realistic and diverse degradations, while DPG generates positive-negative prompt pairs that enhance the diffusion's CFG capability. Extensive experiments demonstrate the effectiveness of HAODiff and the contribution of these modules.

## Acknowledgments

This work was supported by Shanghai Municipal Science and Technology Major Project (2021SHZDZX0102), the Fundamental Research Funds for the Central Universities, the Special Project on Technological Innovation Application for the 15th National Games and the National Paralympic Games under Grant 2025B01W0005.

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
