# OpenReview forum: "HAODiff: Human-Aware One-Step Diffusion via Dual-Prompt Guidance"
_NeurIPS.cc/2025/Conference — NeurIPS 2025 poster_

### Official Review · Reviewer_o2JZ · 2025-06-24

**Clarity:** 3
**Significance:** 3
**Originality:** 3
**Rating:** 4
**Confidence:** 3

**Summary:**

This paper proposes a two-stage framework for human body restoration with a focus on addressing human motion blur. The first stage constructs a realistic degradation pipeline that simulates spatially-aware motion blur using part-level segmentation and morphological operations, enhancing data diversity. The second stage introduces a Dual-Prompt Guidance (DPG) module to extract semantic priors from degraded inputs, which are used to guide a one-step diffusion model for high-fidelity restoration. The methodology is technically solid and well-motivated, combining detailed degradation modeling with efficient generative restoration. However, more analysis on the effectiveness of individual degradation components and a deeper understanding of the prompt embeddings would further strengthen the work.

**Questions:**

- How well does the HMB pipeline generalize to more complex human motion scenarios, such as fast multi-limb movements in sports or dancing with occlusions? Please provide results or analysis on such challenging cases.
- Are there any training stability issues with the three-branch Dual-Prompt setup? Clarify training details and convergence.

**Ethical Concerns:**

["NO or VERY MINOR ethics concerns only"]

**Final Justification:**

The author's response has resolved my doubts, and I will maintain my score. However, I pay little attention to the latest progress in this field. Please also refer to the comments of other reviewers.

**Limitations:**

yes

**Quality:**

3

**Strengths And Weaknesses:**

Strengths:
- The restoration task is well-structured into two clear stages: Dual-Prompt guidance for semantic extraction followed by One-Step Diffusion for efficient detail restoration, effectively addressing the challenge of generating high-quality images from low-quality inputs.
- The use of diffusion inversion with classifier-free guidance, latent VAE sampling, and Performer-based pooling significantly improves sampling efficiency by reducing the need for multiple diffusion steps, resulting in a focused and technically sound approach.

Weaknesses:
- The method’s heavy reliance on precise body part segmentation via Sapiens limits practical applicability, with no discussion on obtaining accurate masks in real-world or unsupervised settings.

- Human motion blur modeling remains overly simplistic, ignoring complexities like varying limb velocities, camera exposure, and depth-of-field effects, which may hinder real-world generalization.

- The Dual-Prompt architecture lacks clear motivation and detailed explanation on why joint training of image, residual, and mask predictions is necessary.

- The approach primarily assembles existing modules (SwinIR, Stable Diffusion, Performer, CFG), limiting novelty to engineering integration.

---

> ### Author Rebuttal · Authors · 2025-07-28
>
> `Q4-1:` The method’s heavy reliance on precise body part segmentation via Sapiens limits practical applicability, with no discussion on obtaining accurate masks in real-world or unsupervised settings.
>
> `A4-1:` Thank you for the insightful comment. We would like to clarify that body part segmentation from Sapiens is **used only during the training phase** of the degradation pipeline and the DPG. It is not used during inference or under unsupervised/real-world settings. During supervised training, the DPG module learns to utilize motion-aware priors derived from segmentation masks. Therefore, actual segmentation results are not required during inference.
>
> Moreover, we do not rely on high-precision segmentation from Sapiens, as motion blur inherently introduces ambiguity in limb boundaries. To better approximate real-world motion blur, we apply morphological operations and Gaussian blurring to the segmentation masks, which helps produce smoother boundary transitions. This enhances the realism of the simulated degradation and helps reduce the domain gap between synthetic and real-world data.
>
> ---
> `Q4-2:` Human motion blur modeling remains overly simplistic, ignoring complexities like varying limb velocities, camera exposure, and depth-of-field effects, which may hinder real-world generalization.
>
> `A4-2:`
> Thank you for highlighting this important aspect. In our degradation pipeline, we adopt the trajectory generation strategy from DeblurGAN [1], which combines inertial motion, Gaussian perturbations (to simulate physiological tremor), and occasional large jitters (to simulate abrupt movements). This results in **diverse and realistic motion trajectories under plausible physical assumptions**.
>
> While we do not explicitly model exposure time or depth-of-field effects, our training data is collected under **diverse real-world conditions that inherently include variations in exposure, lighting, and focus**. We think this enables the model to implicitly learn such effects and remain robust during inference. Moreover, these factors are often entangled in practice and difficult to isolate individually. Thus, we adopt a data-driven approach to better capture their joint impact.
>
> Finally, our proposed MPII-Test dataset, constructed from real-world blurry images, contains diverse and complex motion blur scenarios. Our method achieves state-of-the-art performance on this benchmark, further validating its strong generalization to real-world conditions.
>
> [1] Kupyn et al., DeblurGAN: Blind Motion Deblurring Using Conditional Adversarial Networks, CVPR, 2018.
>
> `Q4-3:` The Dual-Prompt architecture lacks clear motivation and detailed explanation on why joint training of image, residual, and mask predictions is necessary.
>
> `A4-3:`
> Thank you for the question. Our motivation for designing the dual-prompt architecture stems from the observation that current CFG-based image restoration methods typically focus only on the positive prompt, while the negative prompt is fixed or generic. This limits the model’s ability to capture the specific type and region of degradation in the input image. Moreover, vision-language models often struggle to generate precise textual descriptions of degradations, especially in complex or subtle cases. To address these issues, we propose to **jointly learn both positive and negative image-based prompts directly from the degraded input**, enabling more informative and task-specific conditioning.
>
> Instead of using LQ features as the negative prompt, which may retain useful structure, we argue that the residual noise (LQ - HQ) represents more pure degradation signals and is better suited for negative guidance. Conversely, the HQ prediction branch provides high-quality content that aligns with the intended restoration target, serving as a positive prompt. Additionally, we observe that real-world degraded human images often contain localized motion artifacts. To tackle this, we introduce an auxiliary mask prediction branch that captures these local distortions to assist the model in negative prompt modeling.
>
> The overall three-branch structure of the DPG module is inspired by multi-task learning paradigms, where a shared backbone is used to solve multiple related tasks. In our case, the backbone features are passed to three task-specific branches: one for HQ prediction, one for residual noise prediction, and one for HMB mask prediction. These outputs are structurally correlated and jointly optimized, enabling the model to learn richer and more targeted guidance signals while remaining parameter-efficient.
>
>
> `Q4-4:` The approach primarily assembles existing modules (SwinIR, Stable Diffusion, Performer, CFG), limiting novelty to engineering integration.
>
> `A4-4:`
> Thank you for the comment. As noted in our response to `Q2-3`, our work builds upon a one-step diffusion backbone and leverages components such as SwinIR and Stable Diffusion. However, the novelty lies in how these modules are integrated and guided for the specific task of restoring human images degraded by motion blur.
>
> We introduce a new degradation pipeline specifically tailored for HMB, enabling the model to generalize to complex, real-world motion patterns. Moreover, we propose the dual-prompt guidance (DPG) module, which injects both structural and spatial degradation-aware signals into the diffusion process. This enhances the use of classifier-free guidance (CFG) by introducing adaptive negative prompts and localized spatial maps, which are rarely explored in existing frameworks.
>
> While our approach employs existing architectures, it introduces new mechanisms for degradation-aware conditioning and guidance. The consistent improvements across synthetic and real-world benchmarks, including our challenging MPII-Test dataset, demonstrate the practical effectiveness and generalization capability of our framework.
>
> ---
> `Q4-5:` How well does the HMB pipeline generalize to more complex human motion scenarios, such as fast multi-limb movements in sports or dancing with occlusions? Please provide results or analysis on such challenging cases.
>
> `A4-5:`
> Thank you for the valuable question. Our current degradation pipeline is primarily designed for typical scenarios, such as single-person motion or multi-person scenes with limited movement. Although the current pipeline does not explicitly simulate highly dynamic cases, our method has demonstrated good generalization to a wide range of real-world conditions, including moderate motion complexity and partial occlusions.
>
> This question shares a common concern with `Q1-2`, namely that different body parts can undergo motion blur in different directions. As discussed in `A1-2`, our pipeline currently applies a single localized motion trajectory per region, without explicitly modeling joint-level motion differences. Nevertheless, the modular design allows for easy extension to incorporate part-specific motion blur by applying distinct motion kernels to segmented regions, followed by boundary smoothing and image composition. This flexibility provides a practical foundation for handling more complex human motion scenarios.
>
> Extremely dynamic movements involving high-speed multi-limb motion are relatively rare in daily life but can appear more frequently in sports scenes, such as the soccer example shown in Fig. 8. Such cases remain particularly challenging for existing restoration models. We believe that incorporating additional human priors, such as body pose or skeletal information, may be necessary for achieving faithful restoration in these challenging scenarios. We consider it valuable to integrate these complex cases into the degradation process and to investigate targeted restoration strategies.
>
> ---
> `Q4-6:` Are there any training stability issues with the three-branch Dual-Prompt setup? Clarify training details and convergence.
>
> `A4-6:`
> Thank you for the question. Although the DPG adopts a three-branch structure, we did not observe noticeable instability during training. The overall architecture is modular and well-structured, and training proceeds smoothly.
>
> As shown in Fig. 3 Stage 1 and described in Sec. 3.2, the DPG first uses a convolutional head ($H_D$) to downsample the input LQ image by a factor of 4 and increase the channel dimension to match the input format of the Swin Transformer. The shared backbone ($H_E$) then extracts features using two residual Swin Transformer blocks (RSTBs), each with six Swin Transformer layers (STLs). The features are then passed to three branches ($H_{Ri}, i \in [1, 3]$), each with two RSTBs and three STLs, followed by a convolutional upsampling module ($H_{Ui}$). The three branches are used to predict the HQ image, the residual noise, and the HMB segmentation mask.
>
> Training details are described in Sec. 4.1. We employ a joint loss comprising an L1 loss for the HQ image, an L1 loss for the residual noise, and a Dice loss for the segmentation mask. To balance the losses, we set the Dice loss weight to 0.02. We use the Adam optimizer with a learning rate of 2$\times$10$^{-3}$ and a batch size of 16. The model is trained for 20k iterations on four NVIDIA RTX A6000 GPUs. The segmentation branch outputs a single channel with sigmoid activation, while the other two branches output three channels without activation.
>
> Under this setup, the DPG shows stable training in stage 1. We do not observe mode collapse or oscillation. All three loss terms decrease smoothly and converge before the end of training. The loss values at different iterations are shown in the table below.
>
> |Iteration|Total Loss|HQ Loss|Residual Loss|Mask Loss
> |-|-|-|-|-
> |2000|0.1321|0.0600|0.0545|0.0176
> |4000|0.1292|0.0576|0.0539|0.0177
> |6000|0.1260|0.0555|0.0527|0.0178
> |8000|0.1234|0.0540|0.0517|0.0177
> |10000|0.1204|0.0525|0.0505|0.0174
> |12000|0.1192|0.0516|0.0500|0.0176
> |14000|0.1181|0.0512|0.0498|0.0171
> |16000|0.1165|0.0505|0.0492|0.0168
> |18000|0.1161|0.0504|0.0490|0.0166
> |20000|0.1160|0.0506|0.0491|0.0163

---

### Official Review · Reviewer_8mh7 · 2025-07-02

**Clarity:** 3
**Significance:** 3
**Originality:** 3
**Rating:** 4
**Confidence:** 4

**Summary:**

The paper proposes a novel one-step diffusion model, HAODiff, which addresses the challenges of human motion blur and generic degradation in human-centric image restoration through a dual-prompt guidance mechanism. The model leverages a degradation pipeline that simulates realistic degradation scenarios and employs adaptive positive-negative prompts to enhance restoration quality. Experimental results demonstrate its superiority over existing methods.

**Questions:**

see weakness

**Ethical Concerns:**

["NO or VERY MINOR ethics concerns only"]

**Final Justification:**

I incline to accept this paper. The authors address my concerns well in the rebuttal phase.

**Limitations:**

The authors present failure case but lack detailed interpretation and potential solutions.

**Quality:**

3

**Strengths And Weaknesses:**

Strength:
1. The design of the triple-branch DPG mechanism is innovative. It leverages multiple training targets to generate adaptive positive-negative prompts, which enhances the effectiveness of CFG.
2. The design of the degradation pipeline and the dual-prompt guidance mechanism is well-founded and addresses a gap in existing research related to human motion blur. The integration of classifier-free guidance through adaptive positive-negative prompts enhances the model's robustness against diverse degradations.
3. The paper provides extensive experimental results on both synthetic and real-world datasets, including the newly introduced MPII-Test benchmark. The results demonstrate that HAODiff outperforms existing state-of-the-art methods in terms of quantitative metrics and visual quality.
4. The paper is well-organized, with a logical flow from problem statement to methodology, experiments, and discussion.

Weakness:
1. The paper lacks evaluations on classic metric like PSNR, SSIM or MAE, which affects reliability of the model's performance.
2. Fig 8 shows limb poses appear unnatural, and overlapping objects may be misrestored. Can the authors provides the reason and potential solutions?

---

> ### Author Rebuttal · Authors · 2025-07-28
>
> `Q3-1:` The paper lacks evaluations on classic metrics like PSNR, SSIM, or MAE, which affects the reliability of the model's performance.
>
> `A3-1:`
> Thank you for your valuable suggestion. We agree that classic metrics such as PSNR, SSIM, and MAE provide important references for evaluating restoration quality. In our **supplementary Tab. 5**, we report the PSNR and SSIM results for our method and all compared methods, and we additionally include MAE in the table below.
>
> | Metric       | DiffBIR | SeeSR  | SUPIR  | PASD   | ResShift | SinSR  | OSEDiff | InvSR  | OSDHuman | HAODiff |
> |--------------|--------:|-------:|-------:|-------:|---------:|-------:|--------:|-------:|---------:|--------:|
> | **PSNR↑**   | 20.32   | 20.28  | 19.67  | 21.19  | **21.31** | 20.85  | 20.39   | 19.67  | 21.06    | 20.59   |
> | **SSIM↑**   | 0.5592  | 0.5902 | 0.5451 | 0.6248 | **0.6283** | 0.6007 | 0.6049  | 0.5844 | 0.6180   | 0.6035  |
> | **MAE↓**    | 0.0720 | 0.0622 | 0.0654 | **0.0544** | 0.0546 | 0.0570 | 0.0627 | 0.0655 | 0.0566  | 0.0580 |
> | **DISTS↓**  | 0.1402  | 0.1295 | 0.1415 | 0.1469 | 0.1638    | 0.1579 | 0.1340  | 0.1424 | 0.1356   | **0.1023** |
> | **LPIPS↓**  | 0.2797  | 0.2555 | 0.2929 | 0.2910 | 0.2848    | 0.2840 | 0.2507  | 0.2709 | 0.2384   | **0.2046** |
>
> As discussed in **supplementary Sec. C.1 Full-Reference (FR) Metrics**, pixel-level metrics like PSNR often fail to capture perceptual quality in blind restoration. This is because blurry results with oversmoothed textures can still achieve relatively high PSNR and SSIM scores, despite being perceptually inferior. In contrast, perceptual metrics such as DISTS and LPIPS align more closely with human visual preferences. For instance, as shown in **supplementary Fig. 3**, the competing methods produce blurry output that nevertheless receives higher PSNR and SSIM. However, our method achieves significantly sharper restoration and better content fidelity, as reflected in significantly better DISTS and LPIPS scores.
>
> Given the limitations of pixel-level metrics in blind restoration, we focus our discussion in the main paper on perceptual metrics while still providing comprehensive PSNR and SSIM results in the supplementary material to ensure completeness of evaluation.
>
> ---
> `Q3-2:` Fig. 8 shows limb poses appear unnatural, and overlapping objects may be misrestored. Can the authors provide the reason and potential solutions?
>
> `A3-2:`
> Thank you for your insightful comment. In Fig. 8, we believe this failure case arises due to excessive blur in the degraded region and **lacks essential structural cues**. This makes it ambiguous whether the region belongs to a limb or an object. At present, our model relies primarily on visual structure without incorporating explicit structural priors, which limits its ability to restore severely degraded areas.
>
> To address this issue, incorporating structural guidance, such as **pose estimation maps**, could help the model better identify ambiguous regions and generate more anatomically plausible results. Although this strategy is not yet included in our current framework, we believe it is a promising direction for improving restoration performance in future work.

---

> > ### Comment · Reviewer_8mh7 · 2025-08-08
> > **After Rebuttal**
> >
> > Thank you for the response. The rebuttal addresses my concerns.  I agree pixel-level metrics cannot reflect quality of image. I maintain the acceptance of this paper.

---

> > > ### Author Response · Authors · 2025-08-08
> > >
> > > We thank the reviewer for the positive feedback and for acknowledging our rebuttal, and we appreciate the continued support for the acceptance of our work.

---

### Official Review · Reviewer_maZ3 · 2025-07-04

**Clarity:** 3
**Significance:** 2
**Originality:** 2
**Rating:** 4
**Confidence:** 3

**Summary:**

The paper tackles the challenging problem of human body restoration, with a specific focus on handling the co-occurrence of generic image degradation and human motion blur (HMB), a scenario often overlooked by existing methods. The authors propose a comprehensive solution with three main components: A new degradation pipeline that synthetically generates realistic training pairs by first simulating HMB on specific body parts and then applying a sequence of generic degradations. A novel prompt generation module, named Dual-Prompt Guidance (DPG), which is a triple-branch network trained to predict the high-quality (HQ) image, the residual noise (LQ - HQ), and an HMB segmentation mask. These outputs are then used to form adaptive positive and negative prompts for the diffusion model. A one-step diffusion model, HAODiff, that leverages these adaptive prompts via classifier-free guidance (CFG) to achieve efficient and high-fidelity restoration. The authors also introduce MPII-Test, a new benchmark dataset curated for evaluating HBR performance on real-world images with significant motion blur. Extensive experiments show that HAODiff surpasses existing state-of-the-art methods quantitatively and qualitatively.

**Questions:**

Regarding the HMB-R metric, the paper mentions it is calculated using a fine-tuned YOLO model (Sec 4.1), with details in the supplementary material. Given that this is a key metric for validating the central claim of the paper, could the authors provide a brief summary of the detector's performance (e.g., mAP on a hold-out set) in the main paper? The reliability of this metric is crucial for interpreting the results in Table 2.

The DPG architecture (Sec 3.2) features dedicated branches for residual noise and HMB masks. How would this framework generalize to other types of structured degradation, such as heavy rain, snow, or haze? Would it require adding a new, specialized branch for each new degradation type, potentially making the model cumbersome? What are the authors' thoughts on the scalability of the DPG's multi-branch design?

Some other concerns are listed in Weaknesses

**Ethical Concerns:**

["NO or VERY MINOR ethics concerns only"]

**Final Justification:**

The authors' detailed rebuttal has successfully addressed my initial concerns, particularly regarding the two-stage training pipeline and the complexity of the DPG module. I now consider the proposed HAODiff model to be a general and practical approach, and I have raised my final rating accordingly

**Limitations:**

The authors have provided a good discussion of limitations in Section 5, acknowledging failure cases in scenarios with extreme degradation and the modest efficiency penalty from the dual-pass UNet required for CFG. To build on this, I would add:

The paper does not discuss the limitation of its two-stage training pipeline (training DPG, then training OSD), which can be more complex and resource-intensive to manage and tune compared to end-to-end approaches.

As raised in the questions, the current design of DPG with specialized branches for specific degradations might not scale well to a wider variety of real-world distortions, which could be considered a limitation of the current framework's generalizability

**Quality:**

3

**Strengths And Weaknesses:**

Strengths
The paper correctly identifies a significant gap in existing blind image restoration (BIR) and HBR research: the lack of explicit handling for human motion blur, which is a very common artifact. The proposed degradation pipeline (Sec 3.1) is a logical and direct way to address this, and its design, which places HMB in the first-order degradation stage, is physically plausible.

The paper is very well-written and easy to follow. The methodology is explained clearly, and the figures, are illustrative and aid in understanding the proposed framework.

Weaknesses

1. While the guidance mechanism (DPG) is novel, the second stage of the model—the one-step diffusion (OSD) process described in Section 3.3, is largely built upon existing frameworks like OSDHuman and other one-step models. The use of a standard UNet with LoRA, a VAE encoder/decoder, and the standard CFG formulation means the contribution is more about how to guide an existing diffusion framework rather than advancing the diffusion process itself. **This is not a major flaw but slightly reduces the fundamental nature of the contribution.**

2. The proposed solution involves a two-stage training process: first training the DPG module, and then training the OSD model using the frozen DPG as a prompt generator. This introduces **significant complexity** in terms of training procedure and pipeline management compared to end-to-end trainable models.

3. In Section 3.2, the DPG is built with a backbone of two RSTBs (with six Swin Transformer Layers) and three reconstruction branches of another two RSTBs (with three STLs). While this is inspired by SwinIR, it is a rather heavy architecture for a guidance module. It is unclear if this level of complexity is necessary. A lighter backbone might have achieved similar guidance quality while improving overall efficiency.

---

> ### Author Rebuttal · Authors · 2025-07-28
>
> `Q2-1:` Since HMB-R is central to the paper, can the authors briefly report the YOLO detector’s performance (e.g., mAP on a hold-out set) in the main paper to support its reliability?
>
> `A2-1:`
> Thank you for raising this important concern. Since HMB-R is central to evaluating motion blur removal, we agree that clarifying the reliability of the YOLO detector is important. We evaluated our fine-tuned YOLO detector on a dedicated hold-out set of 4,216 images. The evaluation results on this set are presented below to demonstrate its detection quality.
>
> | mAP\@0.5 | Precision | Recall |
> |---------|-----------|--------|
> | 0.6223  | 0.6738    | 0.6110 |
>
> The results show a reasonable level of agreement with the annotated blur regions, suggesting that the detector provides a sufficiently reliable basis for computing the HMB-R metric. In the revised version of the paper, we will provide a summary of these results in the main text to enhance transparency and support the use of HMB-R as a key evaluation metric.
>
> ---
> `Q2-2:` Since DPG uses separate branches for noise and HMB masks, would handling degradations like rain, snow, or haze require adding new branches? Does this affect the scalability and generalizability of the design?
>
> `A2-2:`
> Thank you for the insightful question. The third branch in our DPG architecture was originally designed to predict HMB masks. More broadly, it serves as a general mechanism for providing explicit spatial guidance on localized degradations. Although it is trained on human motion blur in our setting, the underlying principle is applicable to other types of structured degradations such as heavy rain, snow, or haze. These artifacts also exhibit spatial patterns that can be explicitly modeled. **For instance, rain or snow regions can be represented by binary masks, and haze can be described by transmission maps.**
>
> Rather than adding a new branch for each degradation type, the existing mask branch can be extended to **predict multi-channel outputs**, with each channel corresponding to a specific degradation type. For example, one channel may predict the HMB mask, while others predict rain masks or haze transmission maps. This channel-wise design allows the model to handle multiple degradation types simultaneously, without significantly increasing architectural complexity. Therefore, we believe that the current DPG design is both flexible and scalable. The third branch provides a unified and generalizable structure for guiding the restoration process under various types of localized degradation.
>
> ---
> `Q2-3:` Although DPG is novel, the second-stage OSD mainly builds on existing methods, focusing more on guiding a standard diffusion framework rather than advancing it, which slightly weakens the fundamental contribution.
>
> `A2-3:`
> Thank you for the thoughtful comment. In our approach, we choose to retain the existing one-step diffusion (OSD) architecture so as to fully utilize the strong priors in pre-trained diffusion models such as Stable Diffusion. Instead of modifying the model architecture, we focus on developing effective guidance strategies that better adapt these models to realistic and challenging restoration tasks. We believe that enhancing guidance mechanisms can also provide a more scalable and practical path to achieving stronger restoration outcomes.
>
> First, we introduce a new degradation pipeline that explicitly incorporates human motion blur (HMB), enabling the model to learn from more realistic motion patterns and thereby improving generalization to real-world scenarios.
>
> Second, our proposed model, HAODiff, enhances the use of classifier-free guidance (CFG) in one-step diffusion. Previous multi-step methods often use fixed prompts, especially for negative guidance, limiting their effectiveness. Most one-step methods either omit the CFG entirely or set the scale close to 1, making the influence of the negative prompt negligible. In contrast, we adopt adaptive negative prompts in conjunction with a larger CFG scale. This combination effectively pushes the model away from low-quality signals and guides it toward high-quality features, improving both robustness and restoration fidelity.
>
> In addition, we design a dual-prompt guidance (DPG) module that generates both structural features and spatial degradation maps. This allows the model to incorporate not only high-level content information but also explicit localization of degradations such as HMB. Our ablation study demonstrates that injecting spatial information as part of the prompt significantly improves performance, offering a promising new direction for handling localized degradations in diffusion-based restoration models.
>
> The strong performance of our method on both public datasets and our newly introduced benchmark demonstrates the effectiveness of the overall framework, including our degradation simulation strategy, adaptive prompt formulation, and spatial guidance mechanism.
>
> ---
> `Q2-4:` The DPG design is quite heavy for a guidance module. Is such complexity necessary, or could a lighter backbone offer similar performance with better efficiency?
>
> `A2-4:`
> While our method is indeed inspired by SwinIR and adopts Residual Swin Transformer Blocks (RSTBs), we have already introduced **significant architectural simplifications** (e.g., reduced depth and channel size) to reduce the model's size and complexity. Furthermore, compared to other guidance strategies such as SUPIR (which relies on LLaVA for prompt generation and takes several seconds per image), our approach achieves better performance with lower latency. We believe this trade-off is justified by the substantial improvements in guidance effectiveness.
>
> That said, we acknowledge that DPG remains relatively heavy compared to the prompt modules used in other one-step diffusion methods. The table below provides a quantitative comparison between DPG and the full HAODiff model. While DPG contributes minimally to the overall parameter count, it accounts for a relatively high portion of the total MACs. To explore the potential for further simplifying the current architecture, we are currently training a lighter DPG variant to serve as a more efficient guidance module.
>
> |Module|Parameters (M)|MACs (G)
> |-|-|-
> |DPG|38|479
> |Overall|1,459|2,600
>
> ---
> `Q2-5:` The paper does not discuss the limitation of its **two-stage** training pipeline (training DPG, then training OSD), which can be more complex and resource-intensive to manage and tune compared to **end-to-end** approaches.
>
> `A2-5:`
> Thank you for the insightful question. We are fully aware that a two-stage training pipeline introduces additional complexity into the overall framework, as it requires separate optimization of the DPG and OSD components. We did experiment with end-to-end training in the early stages of our work, but the results were suboptimal. We believe this is mainly due to two reasons.
>
> First, jointly training the DPG and OSD introduces multiple competing loss terms, making it difficult to achieve stable convergence. Second, unlike the UNet backbone in Stable Diffusion that benefits from pre-trained priors, the DPG module is trained from scratch and lacks meaningful guidance signals in the early training stages. As a result, it struggles to generate effective prompts for the diffusion model when optimized jointly from the beginning.
>
> To mitigate these issues, we adopt a two-stage training strategy. In the first stage, we train the DPG module separately using the training set as supervision to incorporate prior knowledge into the module. Once trained, the DPG can provide stronger and more consistent guidance during the second-stage training of the diffusion model. While this approach adds some training overhead, we find that it significantly improves both performance and training stability. In the future, we aim to explore unified architectures that enable stable end-to-end training while maintaining or even surpassing the performance of our current HAODiff framework.

---

> > ### Author Response · Authors · 2025-08-04
> >
> > ## **Additional Comment on `Q2-4`**
> >
> > Thank you again for the insightful question. To further address the concern regarding the complexity of the DPG module, we conducted additional experiments by reducing both the number of RSTBs and Performer layers. This resulted in two lighter variants, **DPG-Tiny** and **DPG-Small**, which were evaluated on the PERSONA-Val dataset. The results are summarized below:
> > |Module|Params (M)| MACs (G)|DISTS↓|LPIPS↓|CLIPIQA↑|MANIQA↑
> > |-|-|-|-|-|-|-
> > |DPG-Tiny|27|191|0.1058|0.2134|0.7674|0.7035
> > |DPG-Small|34|420|0.1043|0.2078|0.7642|0.7075
> > |DPG-Base|38|479|**0.1023**|**0.2046**|**0.7737**|**0.7097**
> >
> > These results suggest that while lighter backbones are feasible, reducing the model size leads to noticeable drops in both full-reference (DISTS, LPIPS) and no-reference (CLIPIQA, MANIQA) IQA metrics. This performance degradation highlights the importance of preserving sufficient model capacity within DPG to maintain high-fidelity restoration and semantic consistency. Thus, we believe that the current design of DPG strikes a reasonable balance between efficiency and effectiveness. In future work, we also plan to explore alternative architectural designs that can provide high-quality guidance while further reducing the complexity of the prompt module.
> >
> > ---
> > ## **Further discussion**
> >
> > As the reviewer author discussion phase has now passed the halfway point, we would like to check whether our responses have sufficiently addressed your concerns. If there are any remaining questions or points that need further clarification, we would be happy to provide additional explanation.
> >
> > Thank you in advance for your time and consideration.

---

> > > ### Comment · Reviewer_maZ3 · 2025-08-04
> > >
> > > Thanks for the clarification. My concerns have been solved, and I think that the proposed one-step diffusion model, HAODiff,  is a general and practical approach. Hence, I will raise my final rating.

---

> > > > ### Author Response · Authors · 2025-08-04
> > > >
> > > > Thank you for your kind response. We are very pleased that our clarifications have resolved your concerns, and we will carefully reflect this discussion in the final version of the paper.

---

### Official Review · Reviewer_LbqC · 2025-07-17

**Clarity:** 3
**Significance:** 3
**Originality:** 3
**Rating:** 5
**Confidence:** 4

**Summary:**

The paper proposes a solid solution pipeline for image restoration tasks under human body scene, which consists of a degraded samples generation pipeline, an image restoring pipeline and a human body restoration metrics dataset. The degradation pipeline generate degraded images from high quality images through three paths, which cover the usual image degradation scenes (blur, erosion, dilation, noise, down-sampling, compression), and specially HMB scenes (Human Motion Blur). The restoration pipeline adopts a novel dual-prompt-guidance strategy that split and convert features into positive and negative prompt of CFG. And it outperforms previous methods numerically and visually. Also, the methods is relative light-weight and less performance-consuming, which makes it possible for application on light-weights devices like phones, tables and drones.

**Questions:**

For the sample generation pipeline:

It seems that all the samples finally pass through the compression process (Figure 2). But under the real scene, devices (phones and cameras) usually store uncompressed images. Does the pipeline takes this into consideration and generates samples without compression?

Usually, different parts of human bodies can simultaneously have motion blur in different directions. Does the pipeline takes this into consideration?

Does the pipeline disentangles the motion blur of camera and motion blur of human body, which could happens at same time?

For the restoration pipeline:

How much is the performance gain from adversarial training?

**Ethical Concerns:**

["NO or VERY MINOR ethics concerns only"]

**Final Justification:**

The authors' rebuttal have addressed with my problems. This is a solid, novel and practical work proved with heavy experiments. Also, the method is reasonable, specific-designed while remains the possibility for solving other down-stream tasks. I will keep my rating.

**Limitations:**

The limitations have been thoroughly and clearly discussed in the paper.

**Paper Formatting Concerns:**

All right.

**Quality:**

3

**Strengths And Weaknesses:**

Strengths:

The paper is well organized and the experimental results (metrics and examples) are sufficient to support its claims.

The method achieves SOTA and outperforms previous method from both numerical metrics and visual quality.

The method adopts light-weight modules (e.g. swin-transformer-blocks and performer) and one-step diffusion schema, which makes it fast and less performance-consuming. So, it has the potential availability for inferencing on light-weight devices like phones (which is under the most common scene that users need human blur restoration), tablets and drones.

The proposed strategy dual-prompt guidance is interesting, which logically split and convert additional features into negative and positive prompt of classifier free guidance, has the potential usage for other tasks.

The ablation study is detailed and convincing.

Weakness:

The method is very complex, including training and fine-tuning of multiple models and multiple stages, and also generative adversarial training.

Other: See Questions.

---

> ### Author Rebuttal · Authors · 2025-07-28
>
> `Q1-1:` It seems that all the samples finally pass through the compression process (Figure 2). But under the real scene, devices (phones and cameras) usually store uncompressed images. Does the pipeline take this into consideration and generate samples without compression?
>
> `A1-1:`
> Thank you for the question. In our degradation pipeline, the compression process specifically refers to **JPEG compression**. While high-end applications may store uncompressed RAW images, JPEG remains the dominant format used by most consumer devices, such as smartphones and cameras, due to its balance between image quality and storage efficiency. JPEG is a lossy compression format by design, and compression artifacts exist at all quality levels, including at a quality factor (QF) of 100. Although these artifacts may be imperceptible to the human eye at high quality levels, they are still present.
>
> In our setup, we vary the JPEG quality factor within a range of 30 to 95. A QF of 95 produces images that are nearly indistinguishable from uncompressed versions to the human eye and reflects the common default setting used in many modern devices. A QF of 30 corresponds to scenarios such as extreme compression on social media platforms. This range allows our model to learn robustness across realistic compression levels commonly encountered in real-world scenarios.
>
> ---
> `Q1-2:` Usually, different parts of human bodies can simultaneously have motion blur in different directions. Does the pipeline take this into consideration?
>
> `A1-2:`
> Appreciate the insightful question. In our current degradation pipeline, we do not explicitly simulate distinct motion directions for different human body parts. Instead, we apply a single localized motion trajectory to each selected region. While this design simplifies the motion modeling process, it still produces realistic motion blur patterns that support effective training and generalize well to real-world scenarios.
>
> Importantly, our pipeline is modular and can be readily extended to incorporate part-specific motion blur. For example, during degradation, multiple segmentation masks corresponding to different body parts or subjects can be pre-selected. Distinct motion blur kernels can then be applied to each region individually, followed by boundary smoothing and spatial composition into a coherent degraded image. Although our current implementation does not explicitly simulate fine-grained motion variations across body parts, we recognize its potential to further enhance realism. We consider this a promising direction and plan to incorporate it into our future work.
>
> ---
> `Q1-3:` Does the pipeline disentangle the motion blur of camera and motion blur of human body, which could happen at the same time?
>
> `A1-3:`
> Thank you for raising this important and insightful point. Our current degradation pipeline is primarily designed to simulate localized motion blur caused by human body movement, and does not explicitly disentangle it from camera-induced motion blur, although both types of blur may co-occur in real-world scenarios. We believe that these two types of motion blur can be jointly modeled within a unified framework.
>
> Specifically, motion blur can be interpreted as the temporal integration of pixel intensities over the exposure duration, where each pixel follows a motion trajectory driven by instantaneous velocity. For **camera motion**, a global velocity field is applied uniformly to all pixels. For **human motion**, localized velocity fields are applied only within human regions such as limbs. When both types of motion are present, the per-pixel motion can be approximated as the vector sum of the camera and human velocities, which is then integrated over time to produce the final motion blur kernel.
>
> As a result, different regions in the image can be convolved with kernels derived from different sources: background regions use kernels from camera motion alone, while human regions use combined kernels incorporating both human and camera motion. The final blurry image is generated by fusing the outputs from each region accordingly.
>
> Although our current implementation only simulates human motion blur, this modeling strategy provides a theoretical basis for jointly generating images that contain both types of blur. In future work, we aim to address both concerns raised in `Q1-2` and `Q1-3` within a single pipeline. Specifically, we plan to segment the image into different categories and disconnected regions, and estimate localized motion kernels for each region independently. These region-specific kernels will then be combined with a global camera motion field to simulate the coexistence of camera motion blur and multi-body-part motion blur in a principled and physically consistent manner.
>
>
> ---
> `Q1-4:` How much is the performance gain from adversarial training?
>
> `A1-4:`
> Thank you for the question. The differences between real and generated images are often subtle and difficult to capture using conventional RGB-space loss functions. Prior research [1] shows that these differences are more clearly reflected in the distribution of a pre-trained UNet's output, which provides a promising basis for improving the realism of restored images. Based on this observation, we use an adversarial loss in the latent space, where a discriminator uses the output of the fixed UNet to guide the restored latent vectors toward better alignment with those of real images.
>
> To quantify the contribution of adversarial training, we conduct an ablation study on PERSONA-Val by retraining the model without the GAN loss. As shown in the table below, removing adversarial training results in noticeable performance degradation across both perceptual metrics (DISTS, LPIPS, and TOPIQ) and no-reference IQA scores (CLIPIQA, MANIQA, and NIQE). This confirms that adversarial training significantly enhances the perceptual quality and realism of the restored images.
> |Method|DISTS↓|LPIPS↓|TOPIQ↑|CLIPIQA↑|MANIQA↑|NIQE↓
> |-|-|-|-|-|-|-
> |w/o GAN loss|0.1046|0.2091|0.5106|0.7630|0.7057|3.1745
> |w/ GAN loss|**0.1023**|**0.2046**|**0.5161**|**0.7737**|**0.7097**|**2.8298**
>
> [1] Li, Jianze et al., Unleashing the Power of One-Step Diffusion based Image Super-Resolution via a Large-Scale Diffusion Discriminator, arXiv, 2025.

---

### Author Response · Authors · 2025-08-09
**Response to all reviewers and area chairs for a brief summary**

Dear reviewers and area chairs,

We sincerely thank all reviewers and area chairs for their valuable time, constructive feedback, and encouraging recognition of our work.

We are pleased to note that:
- The novelty and practical value of explicitly addressing **human motion blur (HMB)** alongside generic degradation were recognized across reviewers. For example, LbqC highlighted our efficient one-step design for lightweight devices, while maZ3 and o2JZ acknowledged the realistic degradation pipeline and the adaptive guidance strategy.
- The **dual-prompt guidance (DPG)** mechanism was regarded as innovative by several reviewers, with 8mh7 emphasizing its effective integration of adaptive positive-negative prompts into classifier-free guidance.
- Strong results on both synthetic and real-world datasets, including MPII-Test, were commended; maZ3 explicitly noted the **generality and practicality of HAODiff** after discussion, and 8mh7 maintained support for acceptance.


We have responded to each reviewer individually, and we summarize our key clarifications here:
1. We clarified the rationale behind our **JPEG compression settings**, detailed how the degradation pipeline models HMB, and explained extensions to part-specific and combined camera–human motion blur.
2. We demonstrated the clear perceptual gains from **adversarial training** through ablation studies in the latent space.
3. We reported the mAP, precision, and recall of the YOLO detector to confirm the **reliability of the HMB-R metric**.
4. We explained that the DPG mask branch can output multiple channels for different degradations, **enabling scalability without adding separate branches**.
5. We emphasized our **novelty** in adaptive negative prompts and spatial degradation maps, which enhance CFG effectiveness while leveraging strong pretrained priors.
6. We evaluated **lighter DPG variants**, showing that while feasible, reduced capacity leads to noticeable quality drops; the current design strikes a balance between efficiency and performance.
7. We acknowledged the complexity of the two-stage training but justified it for stability and quality benefits, noting it as a future improvement direction.

Again, we extend our sincere gratitude to all reviewers and area chairs for their thoughtful evaluations and constructive suggestions.

Best regards,
Authors

---

### Decision · Program_Chairs · 2025-09-17

**Decision:**

Accept (poster)

**Comment:**

This paper proposes HAODiff, a one-step diffusion framework for human-centric image restoration, explicitly addressing human motion blur (HMB) alongside generic degradations. The contributions include a realistic degradation pipeline, a Dual-Prompt Guidance mechanism that generates adaptive positive and negative prompts, and a new benchmark for evaluating HMB restoration. The method demonstrates strong performance across synthetic and real-world datasets.

Reviewers appreciated the novelty of handling HMB directly, the innovative DPG design, and the solid experimental validation. Concerns were raised about the complexity of the training pipeline, scalability of the DPG design, reliance on existing diffusion backbones, and missing classical metrics. The rebuttal convincingly addressed these points with additional analysis on JPEG compression, part-specific blur, adversarial training gains, YOLO detector reliability for HMB-R, scalability of the mask branch, lighter DPG variants, and supplementary PSNR/SSIM/MAE results. Reviewers acknowledged these clarifications, with several explicitly raising their scores or confirming acceptance.

Overall, the work is technically solid, novel, and practically relevant, with clear motivation and strong empirical results. I recommend acceptance.